# Significance of Circulating Cell-Free DNA Biomarkers in HBeAg-Negative Chronic Hepatitis B Virus Infection and Their Changes after Treatment Initiation

**DOI:** 10.3390/pathogens12030394

**Published:** 2023-03-01

**Authors:** Nikolaos D. Karakousis, Lampros Chrysavgis, Alkistis Papatheodoridi, Aigli-Ioanna Legaki, Panagiotis Lembessis, Evangelos Cholongitas, Antonios Chatzigeorgiou, George Papatheodoridis

**Affiliations:** 1Department of Gastroenterology, Medical School of National and Kapodistrian University of Athens, General Hospital of Athens “Laiko”, 11527 Athens, Greece; 2Department of Physiology, Medical School of National and Kapodistrian University of Athens, 11527 Athens, Greece; 3Department of Clinical Therapeutics, Medical School of National and Kapodistrian University of Athens, “Alexandra” General Hospital of Athens, 11528 Athens, Greece; 4First Department of Internal Medicine, Medical School of National and Kapodistrian University of Athens, General Hospital of Athens “Laiko”, 11527 Athens, Greece

**Keywords:** hepatitis B, cell-free DNA, 5-methyl-2′-deoxycytidine, global DNA methylation

## Abstract

Background: Chronic hepatitis B virus (HBV) infection is a common chronic liver disease that is closely associated with increased morbidity and mortality. Circulating cell-free DNA (cf-DNA) and global DNA methylation, expressed as circulating levels of 5-methyl-2′-deoxycytidine, are increasingly used to monitor chronic inflammatory diseases of several etiologies. This study attempts to investigate the serum levels of circulating cf-DNA and 5-methyl-2′-deoxycytidine in HBeAg-negative patients with chronic infection (carriers) and chronic hepatitis B (CHB), as well as their changes after treatment initiation in CHB. Methods: Serum samples from a total of 61 HBeAg-negative patients (30 carriers and 31 CHB patients) were included in order to quantify the levels of circulating cf-DNA and 5-methyl-2′-deoxycytidine. In addition, serum samples from 17 CHB patients in complete virological and biochemical remission after initiation of treatment with a nucleos(t)ide analogue were included. Results: Circulating cf-DNA concentration was significantly increased after the initiation of treatment (15 vs. 10 ng/mL, *p* = 0.022). There was a trend in higher mean levels of circulating 5-methyl-2′-deoxycytidine in carriers compared to CHB patients (211.02 vs. 175.66 ng/mL, *p* = 0.089), as well as a trend in increasing 5-methyl-2′-deoxycytidine levels after treatment initiation in CHB patients compared to pre-treatment levels (215 vs. 173 ng/mL, *p* = 0.079). Conclusions: Both circulating levels of cf-DNA and 5-methyl-2′-deoxycytidine might be useful biomarkers in order to monitor liver disease activity and response to antiviral treatment in HBeAg-negative chronic HBV patients, but further studies are essential in order to validate these intriguing findings.

## 1. Introduction

Chronic infection with hepatitis B virus (HBV) is a global public health issue, which can progressively lead to accumulating liver fibrosis, development of cirrhosis and eventually end-stage liver disease [1,2]. HBV also represents a crucial risk factor for the development of hepatocellular carcinoma (HCC) [1]. In recent years, the majority of chronic HBV patients are negative for HBV e antigen (HBeAg) and can be in the inactive phase of chronic infection (carriers) or the active phase of chronic hepatitis B (CHB), which have indications for treatment [1,2]. Current treatment for CHB is mainly based on long-term monotherapy with an oral antiviral agent with a high genetic barrier to resistance [2,3]. During recent decades, antiviral treatment for CHB aims to induce inhibition of HBV replication, which leads to normalization of elevated aminotransferases, amelioration of hepatic necroinflammatory activity and fibrosis, often achieving even reversion of cirrhosis progression and improvement in the patients’ outcome, including survival [1,2,4]. Long-term antiviral treatment decreases the risk of HCC development, but HCC might still develop even after several years of complete inhibition of HBV replication [5,6,7].

Cell-free DNA (cf-DNA) is mostly released from cells that have sustained cell death and apoptosis, but also from other types of living cells, and it mainly consists of free-floating, double-stranded and highly fragmented DNA, with most fragments being approximately 150 bp in length [8,9,10]. Circulating cf-DNA, found in body fluids, can be isolated from serum (liquid biopsy) and includes vital information about several medical conditions [9,11]. Recent studies have suggested that circulating cf-DNA levels can be found elevated in various conditions, such as cancer, chronic inflammation, aging, diabetes mellitus, tissue trauma, sepsis, myocardial infarction or even in physiological nonmalignant conditions, such as those caused by exercise [11,12,13,14]. DNA methylation is an epigenetic alteration that has a pivotal role in various diseases. It occurs with the addition of a methyl group to the fifth carbon of cytosine (5-methylcytosine, 5 mC) via DNA methyltransferases (DNMTs), most frequently at cytosine residues in the sequence context of 5′-C-phosphate-G-3′ (CpG) [15]. DNA methylation can also be detected by “liquid biopsy” and an analysis of the subject’s materials, such as nucleic acids or cells [16]. DNA methylation is a crucial mechanism that controls the proliferation of cells, cell cycle, differentiation, apoptosis and transformation in eukaryotes [17]. Alterations in DNA methylation may be present in aging, cancer and inflammation [17,18,19]. The significance of these biomarkers has been studied in several chronic inflammatory diseases [20,21,22,23], but relevant data in chronic HBV infection are rather limited. Interestingly, we have already reviewed the promising and intriguing interplay among HBV, hepatocellular senescence and apoptosis [24], and this consequently led us to try to further investigate biomarkers related to apoptosis and senescence, such as circulating cf-DNA and DNA methylation as already mentioned, in serum samples from chronic HBV patients.

As a result, our study attempts to determine the levels in the serum of cf-DNA and 5-methyl-2′-deoxycytidine in HBeAg-negative patients with chronic HBV infection at different phases, as well as in CHB patients before and after treatment initiation.

## 2. Materials and Methods

### 2.1. Study Design and Patients

In total, 61 Caucasian adult HbeAg-negative subjects were included in our study. Of them, 30 were classified as cases of chronic infection (carriers) and 31 were classified as patients with CHB. All patients were closely followed at the outpatient liver clinic of the “Laiko” General Hospital of Athens for at least one year before a definite diagnosis of the phase of chronic HBV infection was made. In particular, our patients fulfilled the following inclusion criteria: (a) confirmed HbeAg-negative chronic HBV infection; (b) close follow-up for at least 12 months or until a definite diagnosis of CHB; (c) available stored serum samples; and (d) willingness to provide written informed consent to participate in the study.

The criteria for exclusion were decompensated cirrhosis; HCC; coinfection(s) with hepatitis D, hepatitis C or human immunodeficiency virus; alcohol abuse; history of liver transplantation; and another potential cause of liver injury. In addition, HbeAg-negative patients who could not be clearly classified into the carrier or CHB phase were also excluded.

Informed consent was acquired from all patients included in this study for the anonymous use of their data and biological material. This study was conducted according to the Declaration of Helsinki and was approved by the Ethics Committee of the General Hospital of Athens “Laiko” and of the Medical School of National and Kapodistrian University of Athens.

### 2.2. Definitions

The diagnosis of HbeAg-negative chronic HBV infection was based on positive HbsAg and negative HbeAg for ≥6 months. HbeAg-negative CHB was diagnosed in HbeAg-negative patients with elevated alanine aminotransferase (ALT) of at least twice the upper limit of normal (ULN), serum HBV DNA > 20,000 IU/mL or serum HBV DNA > 2000 IU/mL, and at least moderate liver necroinflammation and/or fibrosis. HBeAg-negative chronic HBV infection, which is also known as the “carrier” phase, was diagnosed in HBeAg-negative patients with undetectable or low (<2000 IU/mL) serum HBV DNA levels and persistently normal ALT levels according to the traditional cut-off values, during at least four 3-monthly determinations within 12 months. The diagnosis of cirrhosis (always compensated) was validated with compatible histological, ultrasonographic and/or endoscopic findings.

### 2.3. Follow-Up

All subjects of our study were followed according to clinical practice based on local guidelines. Both physical examination and laboratory tests, including hematological and biochemical parameters, were obtained at least every 3–6 months, along with serum HBV DNA levels, which were determined every 6–12 months according to the phase of infection and the use of oral antiviral treatment. The surveillance of HCC was based on 6-monthly ultrasound imaging.

No patient was receiving any type of antiviral at the onset of follow-up. Demographic data concerning age, gender, height, weight, smoking habits, alcohol use, chronic or occasional drug use and biochemical parameters, such as liver function tests (namely ALT, aspartate aminotransferase (AST), alkaline phosphatase (ALP), gamma-glutamyl transferase (GGT) and fasting glucose) and other patient characteristics were recorded at the baseline visit. Moreover, serum samples were collected at the baseline visit. Additionally, in 17 of the 31 CHB patients who started oral antiviral treatment, serum samples were also obtained at least 6 months or more after treatment initiation. All 17 patients achieved complete virological and biochemical remission under monotherapy with a nucleos(t)ide analogue (entecavir or tenofovir disoproxil fumarate) and all on-treatment serum samples were obtained after the induction of remission.

### 2.4. Laboratory Analysis

All blood samples were centrifuged at 1500× *g* for 15 min whilst the serum layer was cautiously transferred to a fresh polypropylene tube and stored at −80 °C until analysis. After equilibration of the stored samples of serum initially at 4 °C and thereupon to room temperature (RT), further centrifugation of our samples was executed at 400× *g* for 2 min as a way to remove any potential remaining components of cells.

The “Plasma/Serum Cell-Free Circulating DNA Purification Mini Kit (Cat. No 55100Dx)” (Norgen Biotek. Corporation, CANADA) was used in order to isolate circulating cf-DNA from 200 μL serum samples [25,26]. For every sample, the concentration of the extracted circulating cf-DNA was determined by quantitative real-time polymerase chain reaction (RT-qPCR) assay, using glyceraldehyde 3-phosphate dehydrogenase (GAPDH) as the housekeeping gene. The relevant primers utilized in our study were GAPDHf: 5’-GGAAGGTGAAGGTCGGAGTC-3 and GAPDHr: 5’ GAAGATGGTGATGGGATTTC-3 [25,26]. All of our samples were analyzed in duplicate. The final value used was derived from the mean of the two measurements. The iTaq™ Universal SYBR Green^®^ Supermix (BIORAD cat no. 172-5124) was used to perform the qPCR reactions, while the human “TaqMan Control Genomic DNA” (Applied Biosystems™ Cat No. 4312660, Waltham, MA, USA) was utilized in order to obtain the standard curve of the exact circulating cf-DNA concentrations of the measured samples [27].

SaCycler-96 (Sacace Biotechnologies) with 40 cycles of amplification under standard cycling conditions (95 °C for 4 min, 95 °C for 15 s, 60 °C for 30 s) was utilized in order to conduct the quantitative PCR assays. Melting point analysis at 65–95 °C was conducted after the amplification reaction, whilst on each PCR plate non-template, the controls were included in order to exclude the possibility of contamination.

As an indicator of DNA methylation, the circulating levels of 5-methyl-2′-deoxycytidine, were quantified by using the samples of serum, along with the “DNA Methylation Elisa Kit” (Cayman Chemical Cat No. 589324, Ann Arbor, MI, USA), according to the manufacturer’s directions [25].

### 2.5. Statistical Analysis

Statistical analysis was performed utilizing SPSS Statistics for Windows (Version 28.0, IBM Corp Armonk, Armonk, NY, USA, 2021) and Graph Pad Prism (GraphPad Software 8, Inc., San Diego, CA, USA). All quantitative variables were presented as mean ± standard deviation (SD) or median values {interquartile range (IQR)} depending on the distribution of each variable, which was evaluated using the Shapiro–Wilk test. The comparison of parametric and non-parametric quantitative variables between two independent groups was conducted using Student’s t-test and the non-parametric Mann–Whitney test, respectively. The comparison of parametric and non-parametric quantitative variables between two related groups was conducted utilizing paired t-test or Wilcoxon signed-rank test, respectively. We used the maximum available data of each parameter for the aforementioned statistical tests. The ROUT method for the identification of outliers was conducted for the analysis of each quantitative variable and, during DNA methylation analysis, one outlier was detected and subsequently removed. The associations between quantitative variables were evaluated using Spearman’s correlation and expressed by Spearman’s coefficient (r). Categorical variables were summarized as frequencies and percentages and their associations were determined utilizing corrected chi-squared or two-sided Fisher’s exact test. Statistical significance was set at *p* value < 0.05.

## 3. Results

### 3.1. Patient Characteristics

The main baseline characteristics of the 61 patients are summarized in Table 1. CHB patients compared to carriers had significantly older mean age (49 vs. 42 years, *p* = 0.033) and significantly higher median serum levels of HBV DNA (314,000 vs. 229 IU/mL), ALT (95 vs. 20 IU/L), AST (50 vs. 18 IU/L) (all *p* < 0.001), GGT (27 vs. 17 IU/L, *p* = 0.001) and ALP (94 vs. 72 IU/L, *p* = 0.015). In contrast, these two groups did not differ significantly in other demographic and behavioral parameters, such as gender distribution, alcohol consumption and smoking habits, or in other laboratory parameters, such as concentration of total protein, albumin and platelet counts, or in the presence of comorbidities, such as type 2 diabetes mellitus (T2DM), dyslipidemia or other chronic diseases. Of the 31 CHB patients, 7 were diagnosed with compensated cirrhosis.

The changes in basic laboratory parameters after the initiation of oral antiviral treatment and, in particular, on the day of the second serum sample in the 17 CHB patients are presented in Table 2. At the time of the second serum sample, all 17 patients had undetectable serum HBV DNA. They had achieved significantly lower and always normal ALT, AST and GGT levels, as well as significantly lower ALP levels, compared to their pre-treatment values.

Results concerning circulating cf-DNA could be obtained from the baseline serum samples of all 61 patients, as well as from the 17 on-treatment serum samples of the 17 treated CHB patients. In our global DNA methylation analysis, results could be obtained from 54 of the 61 baseline serum samples [of 27 carriers and 27 CHB patients (including 5 of the 7 CHB patients with compensated cirrhosis)], as well as from 16 of the 17 on-treatment serum samples of the 17 treated CHB patients.

### 3.2. Circulating cf-DNA Species

No significant difference was observed in the circulating cf-DNA concentration between carriers and CHB subjects (11.3 vs. 13.0 ng/mL, respectively, *p* = 0.559). On the other hand, it seemed that treatment for CHB had an impact on the serum abundance of circulating cf-DNA, as evaluated in 17 patients with serum available before and after CHB treatment initiation. Compared to baseline, circulating cf-DNA concentration was significantly increased after the initiation of treatment (10 vs. 15 ng/mL, respectively, *p* = 0.022) (Figure 1a). No remarkable difference was identified in serum cfDNA concentration between carriers and CHB patients after treatment (11.3 vs. 15 ng/mL, *p* = 0.173). Moreover, the stage of cirrhosis did not seem to affect any of these markers as there was no significant difference in the serum levels of cf-DNA between the 7 CHB patients with cirrhosis and the 24 non-cirrhotic CHB patients (8 vs. 13 ng/mL, respectively, *p* = 0.199) (Figure 1b), or in the levels of 5-methyl-2′-deoxycytidine between 5 CHB subjects with cirrhosis and 22 non-cirrhotic CHB subjects (204 vs. 169 ng/mL, respectively, *p* = 0.336).

Furthermore, there was a trend towards higher mean levels of circulating 5-methyl-2′-deoxycytidine, as a surrogate marker of global methylation of the DNA, in carriers compared to CHB patients (211 vs. 176 ng/mL, respectively, *p* = 0.089) (Figure 2a)**,** as well as an inclination for increased levels of circulating 5-methyl-2′-deoxycytidine in CHB patients after treatment initiation compared to their pre-therapy levels (215 vs. 173 ng/mL, respectively, *p* = 0.079) (Figure 2b). Similar to cfDNA levels, no significant alteration in the mean concentration of 5-methyl-2′-deoxycytidine was observed between carriers and CHB patients after treatment initiation (211 vs. 214.8 ng/mL).

No significant correlation between the levels of circulating cf-DNA or DNA methylation and the patients’ epidemiological or clinical characteristics, except from significantly higher levels of 5-methyl-2’-deoxycytidine in male patients compared to female patients (*p* = 0.001) (Table 3), was identified. Moreover, no significant correlation between the serum levels of cf-DNA and the levels of serum HBV DNA, or any other laboratory parameter in all serum samples from all our CHB subjects, was observed, as well as in the samples from carriers or the 17 samples drawn during treatment. Only in the samples of CHB patients before treatment initiation was there a trend in the correlation between serum levels of DNA methylation with levels of AST (r = 0.372, *p* = 0.06), ALT (r = 0.318, *p* = 0.10), GGT (r = 0.330, *p* = 0.09) and ALP (r = 0.403, *p* = 0.05).

## 4. Discussion

This is the first study based on the liquid biopsy approach that attempts to evaluate the significance of circulating cf-DNA species in the two phases of HbeAg-negative chronic HBV infection, along with the changes in these biomarkers after treatment initiation in CHB patients [28,29]. In particular, we studied the levels of circulating cf-DNA and 5-methyl-2′-deoxycytidine reflecting global DNA methylation, in well-characterized subjects with HBeAg-negative chronic infection (carriers) or CHB. In addition, these biomarkers were also assessed before and after treatment initiation in a subset of our patients with HBeAg-negative CHB.

Ιt is well established that CHB is associated with increased HBV replication and viral load, elevated markers of hepatocyte injury (ALT/AST) and persistent necroinflammatory activity with accumulating fibrosis, while subjects with HbeAg-negative chronic infection (carriers) have low or no HBV replication, normal ALT/AST and an absence of ongoing necroinflammatory activity [1,2]. Inflammation is generally linked to the induction of DNA damage, mutations and cancer progression [30], whilst damaged cells, which are candidates for malignant transformation, can be neutralized by either apoptosis or the senescence pathway [31,32].

Cf-DNA consists of small nucleic acid fragments mostly deriving from apoptotic cells [10,33]. Cell-cycle arrest, cellular senescence and apoptosis can be particularly induced by DNA damage, in general [34]. As a result, the increased inflammation-induced DNA damage in infected hepatocytes in CHB patients compared to carriers could lead to either hepatocellular apoptosis, probably combined with increased circulating cf-DNA levels as an apoptosis product, or the activation of the senescence pathway in order to avoid HCC progression [10,31,32,35]. In our study, we did not identify any significant difference in the levels of circulating cf-DNA between carriers and CHB patients under no treatment. The absence of higher circulating cf-DNA levels, which can be a product of cellular apoptosis [10], in the serum samples of our CHB subjects compared to our carriers seems to favor the establishment of a senescent state among their infected and damaged hepatocytes rather than an apoptosis scenario, as hepatocellular senescence might antagonize apoptosis in CHB.

Interestingly, we found a trend in greater global DNA methylation, expressed as higher circulating levels of 5-methyl-2′-deoxycytidine, in our carriers compared to CHB patients. It has been already recorded that global DNA hypomethylation is associated with cellular senescence, whilst the hepatitis B virus X (HBx) protein seems to promote both specific regional hypermethylation and global hypomethylation [36,37]. These data could also be indicative of a senescence status concerning the hepatocytes of our CHB patients.

The main outcome of this study was the significant increase in the levels of circulating cf-DNA during antiviral treatment in our CHB patients, compared to pre-treatment levels. Treatment with nucleos(t)ide analogues, which is the standard of care for CHB and was used in our patients, has been shown to generally affect DNA structural integrity, leading to stalled replication forks, chain termination and activation of checkpoint pathways, which halt cell-cycle progression [38]. Moreover, nucleos(t)ide analogues activate signaling pathways that initiate the process of apoptosis, which could be responsible for the higher levels of circulating cf-DNA after nucleos(t)ide analogue use in our CHB patients [10,38].

The circulating levels of 5-methyl-2′-deoxycytidine (global DNA methylation) were recorded to have a trend in increase after treatment with nucleos(t)ide analogue in our CHB subjects, compared to their pre-treatment levels. DNA methylation has been previously demonstrated to be related to transcriptional repression [39,40], whereas nucleos(t)ide analogues interfere with endogenous nucleotide metabolism and suppress viral replication [41]. As such, the trend in increasing circulating levels of 5-methyl-2′-deoxycytidine after treatment initiation in our CHB patients could be due to the suppression of viral replication. Lastly, we did not observe any significant difference in the serum abundance of cfDNA and 5-methyl-2′-deoxycytidine between carriers and CHB subjects after treatment, a finding that might be attributed to the fact that the biochemical parameters reflecting inflammation seem similar in these two groups and the viral load is low [2].

Our study has several limitations. Most significantly, this was a single-center study associated with a relatively limited number of patients that can neither provide strong positive findings or exclude type II errors of non-significant results. Given the limited number of patients and samples especially after treatment, the significance of cfDNA markers in subgroups of patients could not be reliably determined. Therefore, larger cohorts are required in order to further assess our findings and illuminate the potential role of these biomarkers.

In conclusion, both circulating levels of cf-DNA and 5-methyl-2′-deoxycytidine might be useful biomarkers for monitoring liver disease activity and response to antiviral therapy in HBeAg-negative chronic HBV subjects. Our study represents a preliminary effort for the understanding of the significance of serum circulating cf-DNA biomarkers in the natural course of HBeAg-negative chronic HBV infection, as well as of their changes after treatment initiation in such CHB subjects, but more studies are required in order to corroborate our findings.

## Figures and Tables

**Figure 1 pathogens-12-00394-f001:**
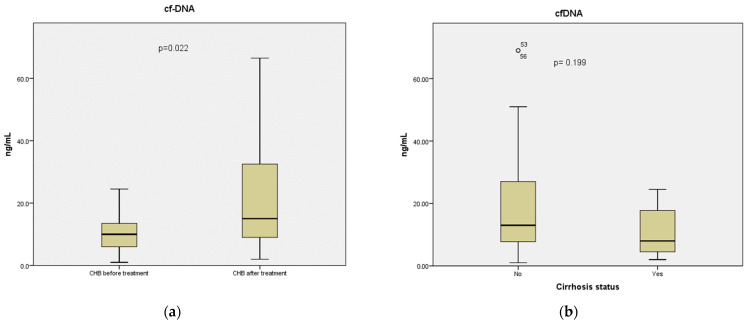
Levels of circulating cell-free DNA (cf-DNA) (ng/mL) in serum samples obtained from (**a**) 17 patients with HBeAg-negative chronic hepatitis B before and after initiation of effective antiviral treatment and (**b**) 7 HBeAg-negative CHB patients with compensated cirrhosis and 24 such patients without cirrhosis. Data are presented in box and whisker plots, which demonstrate median values and 25th–75th percentiles.

**Figure 2 pathogens-12-00394-f002:**
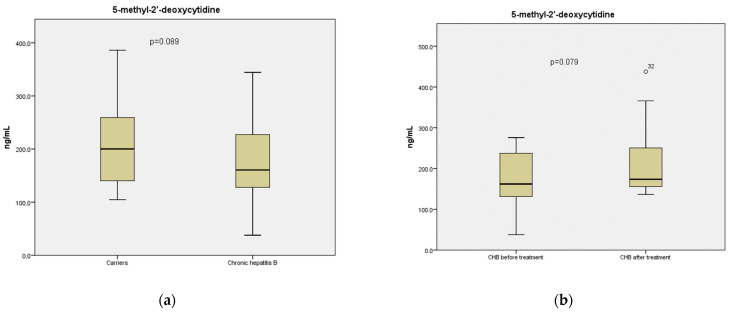
Levels of circulating 5-methyl-2′-deoxycytidine concentration (ng/mL) in serum samples obtained from (**a**) 27 subjects with HBeAg-negative chronic HBV infection (carriers) and 27 patients with HBeAg-negative chronic hepatitis B, and (**b**) 16 patients with HBeAg-negative chronic hepatitis B before and after initiation of effective oral antiviral treatment. Data are demonstrated in box and whisker plots, which present median values and 25th–75th percentiles.

**Table 1 pathogens-12-00394-t001:** Main characteristics of 30 patients with HBeAg-negative chronic hepatitis B virus (HBV) infection (carriers) and 31 patients with HBeAg-negative chronic hepatitis B (CHB).

	HBV Carriers (*n* = 30)	HBeAg-Negative CHB Patients (*n* = 31)	*p* Value
Age, years	42 ± 11	49 ± 13	0.033
Sex, males (%)	18 (60)	17 (55)	0.684
Alcohol, *n* (%) No use Mild use	21 (70) 9 (30)	24 (77) 7 (23)	0.510
Smoking, *n* (%) No Yes	21 (70) 9 (30)	25 (81) 6 (19)	0.334
Type 2 diabetes, *n* (%) No Yes	29 (96.7) 1 (3.3)	30 (96.8) 1 (3.2)	0.981
Dyslipidaemia, *n* (%) No Yes	28 (93.3) 2 (6.7)	30 (96.8) 1 (3.2)	0.534
Other comorbidities, *n* (%) No Yes	24 (80) 6 (20)	22 (71) 9 (29)	0.413
HBVDNA (IU/mL)	229 {556}	314,000 {3,490,000}	<0.001
ALT (IU/L)	20 {13}	95 {102}	<0.001
AST (IU/L)	18 {6}	50 {71}	<0.001
ALP (IU/L)	72 {50}	94 {75}	0.015
GGT (IU/L)	17 {14}	27 {32}	0.001
Total protein (g/L)	71.5 ± 4	72 ± 4	0.660
Albumin (g/L)	44 ± 4	42 ± 3	0.090
Platelets (×10^9^/L)	204 {72}	176 {81}	0.169

Quantitative variables are presented as mean values ± standard deviation or median values {interquartile range}. ALT: alanine aminotransferase; AST: aspartate aminotransferase; ALP: alkaline phosphatase; GGT: gamma glutamyl-transferase.

**Table 2 pathogens-12-00394-t002:** Main characteristics of 17 patients with HBeAg-negative chronic hepatis B (CHB) before and after initiation of oral antiviral treatment.

	before Treatment	on Treatment	*p* Value
ALT (IU/L)	71 {100}	25 {12}	<0.001
AST (IU/L)	50 {59}	24 {8}	0.001
ALP (IU/L)	96 {107}	73 {18}	0.008
GGT (IU/L)	30 {28}	16 {4}	0.002
Total protein (g/L)	72 ± 4	72 ± 3	0.609
Albumin (g/L)	42 ± 3	41.5 ± 3	0.668
Platelets (×10^9^/L)	194 {79}	170 {85}	0.776

Quantitative variables are presented as mean values ± standard deviation or median values {interquartile range}. ALT: alanine aminotransferase; AST: aspartate aminotransferase; ALP: alkaline phosphatase; GGT: gamma glutamyl-transferase.

**Table 3 pathogens-12-00394-t003:** Associations between the circulating cell-free DNA (cf-DNA) and 5-methyl-2′-deoxycytidine levels with clinical and biochemical characteristics of 61 patients with HBeAg-negative chronic HBV infection (carriers: *n* = 30, chronic hepatitis B: *n* = 31).

	cf-DNA		5-methyl-2′-deoxycytidine	
Age, years	r = −0.030	*p* = 0.816	R= −0.018	*p* = 0.896
Sex, male female	13.0 {16.0} 11.3 {15.5}	*p* = 0.688	216.3 {126.6} 135.0 {85.2}	***p* = 0.001**
Alcohol, No use Mild use	13.0 {16.8} 8.5 {12.3}	*p* = 0.099	183.3 {114.1} 189.5 {148.0}	*p* = 0.479
Smoking, No Yes	13.3 {18.3} 9.5 {9.5}	*p* = 0.110	168.8 {116.8} 192.1 {126.6}	*p* = 0.221
HBVDNA (IU/mL)	r = 0.143	*p* = 0.274	r = −0.134	*p* = 0.333
ALT (IU/L)	r = 0.113	*p* = 0.385	r = −0.057	*p* = 0.680
AST (IU/L)	r = 0.083	*p* = 0.525	r = −0.072	*p* = 0.606
ALP (IU/L)	r = 0.022	*p* = 0.869	r = −0.061	*p* = 0.671
GGT (IU/L)	r = 0.033	*p* = 0.804	r = 0.088	*p* = 0.532
Total protein (g/L)	r = 0.023	*p* = 0.894	r = −0.034	*p* = 0.854
Albumin (g/L)	r = −0.088	*p* = 0.615	r = 0.012	*p* = 0.948
Platelets (×10^9^/L)	r = 0.155	*p* = 0.238	r = −0.040	*p* = 0.775

## Data Availability

All data generated or analyzed during this research are included in this published article.

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
