# Peer review of "Significance of Circulating Cell-Free DNA Biomarkers in HBeAg-Negative Chronic Hepatitis B Virus Infection and Their Changes after Treatment Initiation"

_pathogens, 2023, doi:10.3390/pathogens12030394_

Round 1

Author Response

# Reviewer 1

In this manuscript, the Authors aimed to determine the serum levels of cf-DNA and 5-methyl-2’-deoxycytidine in HBeAg-negative patients with chronic HBV infection at different phases as well as in CHB patients before and after treatment initiation. The Authors referred to several studies showing that both circulating cf-DNA levels and alterations in DNA methylation can be found elevated in various chronic inflammation conditions. However, they found that:

- There was a trend for higher mean levels of circulating 5-methyl-2’-deoxycytidine in carriers compared to CHB patients

- Circulating cf-DNA concentration was significantly increased after treatment initiation

- There was a trend for increase of 5-methyl-2’-deoxycytidine levels after treatment initiation in CHB patients compared to pre-treatment levels

The paper is very well written however the findings are not easy to interpret.

Major comments:

The results of the study are not easy to interpret. HBeAg-negative chronic HBV infection or “carrier” phase is characterized by persistently undetectable or low (< 2000 IU/mL) serum HBV DNA levels and persistently normal ALT levels. In this phase of immune control of infection, there is no or minimal liver inflammation, therefore one would expect that in inactive carriers the levels of biomarkers of inflammation would be lower than in patients with chronic hepatitis B. In addition, in CHB patients long-term nucleos(t)ide analogs therapy induce inhibition of HBV replication, which leads to normalization of elevated aminotransferases, amelioration of hepatic necroinflammatory activity and fibrosis often achieving even reversion of cirrhosis. One would expect that after treatment initiation and biochemical and virological response, the levels of biomarkers of inflammation would be lower compared to pre-treatment levels.

Could the Authors better explain their results?

Response: Thank you very much for your intriguing comment. According to current literature, these biomarkers are not specific inflammation markers, but they are also associated with epigenetic changes which are the result of the inflammation effect.

It is recorded that inflammation can cause DNA damage to cells and DNA damage is associated with cell cycle arrest that leads either in apoptosis or in senescence. CHB patients may sustain, because of increased inflammation state, more hepatocellular DNA damage than carriers, and as a result more increased rate of cell cycle arrest and either increased apoptosis or senescence. More apoptosis possibly means higher circulating cell-free DNA levels. Considering that DNA damage is higher at CHB patients and cell cycle arrest is more frequent, we would expect increased cf-DNA levels, if apoptosis took place, but the levels are not significantly increased in comparison with the carriers in our study, despite the increased inflammation rate and cell cycle arrest. As a result, this might mean that the hepatocytes followed the other pathway of cell cycle arrest which is senescence. Τhis hypothesis is also supported by the global hypomethylation state, which was demonstrated in our CHB patients, and is associated with senescence, as it has already been reported and cited in our manuscript.

Concerning CHB patients before and after treatment, it is established by studies already cited in our manuscript that the antiviral treatment in CHB can cause cell cycle arrest, which is known to lead to either apoptosis or senescence. As a result, the increased levels of cell free DNA, which is mainly an apoptosis product, might be the result of apoptosis.

In addition, concerning CHB patients before and after treatment, the trend for increased levels of DNA methylation in patients after treatment, might be associated with the fact that viral replication is suppressed by antiviral drugs against hepatitis B and transcriptional repression is also associated with DNA methylation as cited in our manuscript.

Minor comments

In table 1 and in the text, the comorbidities should be detail

Response: We thank the reviewer for his/her positive remarks and the helpful critique to improve our manuscript. Comorbidities are recorded more thoroughly in table 1 and in text of our revised manuscript.

Reviewer 2 Report

This study was conducted to evaluate the serum levels of circulating cell-free DNA and global DNA methylation in HBeAg-negative patients with chronic HBV infection and chronic hepatitis B, before and after NA treatment. This study may provide certain evidence on the significance of circulating cell-free DNA as potential biomarkers in the course of chronic HBV infection. However, there are some concerns that will need to be addressed.

Specific Comments:

1. The main concern is that the data were not presented sufficiently and properly. Only the comparisons of cf-DNA and 5-methyl-2-deoxycytidine levels in different groups were showed as figures in the manuscript. The data of correlation analysis between cf-DNA and 5-methyl-2-deoxycytidine and other clinical and virological parameters were not presented, thought the results were described briefly. Besides, the data on whether cirrhosis status affect the cf-DNA level should be also presented and included in the current Figure 1. The current Figure 2 and Figure 3 can be grouped into one single Figure.

2. Comparisons and analysis of cf-DNA and 5-methyl-2-deoxycytidine levels in carriers and CHB after NA treatment may be included in the description and discussion in this manuscript, since the biochemical parameters reflecting inflammations seems similar in these two groups.

Author Response

#Reviewer 2

This study was conducted to evaluate the serum levels of circulating cell-free DNA and global DNA methylation in HBeAg-negative patients with chronic HBV infection and chronic hepatitis B, before and after NA treatment. This study may provide certain evidence on the significance of circulating cell-free DNA as potential biomarkers in the course of chronic HBV infection. However, there are some concerns that will need to be addressed.

Specific Comments:

  1. The main concern is that the data were not presented sufficiently and properly. Only the comparisons of cf-DNA and 5-methyl-2′-deoxycytidine levels in different groups were showed as figures in the manuscript. The data of correlation analysis between cf-DNA and 5-methyl-2′-deoxycytidine and other clinical and virological parameters were not presented, thought the results were described briefly. Besides, the data on whether cirrhosis status affect the cf-DNA level should be also presented and included in the current Figure 1. The current Figure 2 and Figure 3 can be grouped into one single Figure.

Response: We thank the reviewer for these useful suggestions that helped improve our manuscript. To address these comments, we have added a table concerning correlations of these biomarkers with other clinical and virological parameters in our revised manuscript. Moreover, the addition of another figure demonstrating whether cirrhosis status affect the cf-DNA level has been added in our revised manuscript along with the appropriate arrangements concerning figures structure.

  1. Comparisons and analysis of cf-DNA and 5-methyl-2′-deoxycytidine levels in carriers and CHB after NA treatment may be included in the description and discussion in this manuscript, since the biochemical parameters reflecting inflammations seems similar in these two groups.

Response: We thank the reviewer for this comment. In our revised manuscript, a figure concerning comparisons and analysis of cf-DNA and 5-methyl-2′-deoxycytidine levels in carriers and CHB after NA treatment has been added along with the appropriate discussion into our text.

Reviewer 3 Report

The manuscript by Karakounis et al. is the first study which investigates the significance of circulating cell-free DNA and global DNA methylation in HBeAg-Negative Chronic HVB infection and their changes after initial treatment by liquid biopsy approach. The manuscript is well-written and easy to follow. The methods are generally appropriate and clearly described. The results are clearly explained while the discussion contain the arguments supported by the experimental data. However, the introduction merely includes a simple review about HBV, cell-free DNA, and DNA methylation without the motivation and the importance of this study. Therefore, I recommend publishing this manuscript with an addition of the aforementioned parts. A short review about state of the art also can improve its quality.

Author Response

#Reviewer 3

The manuscript by Karakounis et al. is the first study which investigates the significance of circulating cell-free DNA and global DNA methylation in HBeAg-Negative Chronic HVB infection and their changes after initial treatment by liquid biopsy approach. The manuscript is well-written and easy to follow. The methods are generally appropriate and clearly described. The results are clearly explained while the discussion contain the arguments supported by the experimental data. However, the introduction merely includes a simple review about HBV, cell-free DNA, and DNA methylation without the motivation and the importance of this study. Therefore, I recommend publishing this manuscript with an addition of the aforementioned parts. A short review about state of the art also can improve its quality.

Response: Thank you for your positive comment. In our revised manuscript, we extended the introduction with further explanations about the motivation and the importance of our study important.

Round 2

Reviewer 1 Report

I feel that this revised version of the manuscript can be accepted for publication.

Reviewer 2 Report

All concerns have been addressed in the revised manuscript.